# Rapid Assessment Method for Evaluation of the Weighted Contribution of Anthropogenic Pollution: A Case Study of Lake Burullus, Egypt

**M. S. Moussa [1,2] and Mohamed K. Mostafa [3,*]**

1    Faculty of Engineering Mataria, Helwan University, Cairo 11718, Egypt; mbadawy@zewailcity.edu.eg
2    Environmental Engineering Program, Zewail City of Science and Technology, 6th October, Giza 12578, Egypt
3    Faculty of Engineering and Technology, Badr University in Cairo (BUC), Cairo 11829, Egypt
*    Correspondence: m_khaled@buc.edu.eg

**Abstract:** This paper proposes a pragmatic approach for rapid assessment of the weighted contribution of the main waste streams contributing to pollution of surface water bodies. A case study was conducted on Lake Burullus in Egypt. The lake suffers from pollution due to many human-based activities around the lake, such as domestic, industrial, agriculture, fish farming, and solid wastes. The weighted contribution of these activities was assessed in terms of chemical oxygen demand (COD), total nitrogen (TN), and total phosphorus (TP). The results showed that the highest organic load is due to the domestic wastewater pollution component (63.2% of COD load), followed by fish aquaculture (35.4%). The highest TN (43.9%) and TP (58.3%) pollutant loads to Lake Burullus are due to the agricultural pollution component, followed by fish aquaculture with pollutant loads of 28.5% and 25.3%, respectively. The industrial wastewater pollution component has a very small effect on the pollution of Lake Burullus. The assessment of this study will help identify and magnify the key polluters and thus guide the decision-makers to prioritize investment planning for depollution intervention projects. For example, if the target is to reduce COD loads, investments must be directed toward the rehabilitation and expansion of wastewater treatment plants (WWTPs).

**Keywords:** anthropogenic activities; chemical oxygen demand; Lake Burullus; Egypt; total nitrogen; total phosphorus; pragmatic approach

## 1. Introduction

Rapid population growth and urban development around freshwater rivers and deltas contribute to the increased pollution load to downstream lakes and finally coastal areas [1,2]. Nearly 400% of the world population lives within 100 km of a coastline [3]. Additionally, many mega-cities are situated in coastal areas [4]. There has been a dramatic increase in human-based activities as a result of population growth to fulfill economic development needs in different sectors (agriculture, industrial, urban development business, fish farming, etc.) [5–7]. The impact of such diverse human-based activities clearly manifests at downstream rivers, lakes, and coastal zones in the form of pollutants and environmental asset deterioration [2]. Industrial wastewater is the main source of heavy metals in surface water, while untreated domestic wastewater is the main source of organic compounds [8,9]. Devlin et al. [10] have reported a deterioration in the water quality of the Pacific coastal waters due to the increase in agricultural runoff, and industrial and urban sewage discharges. Tanjung et al. [11] have reported an increase in the concentration of some water parameters such as nitrate ($NO_3$), phosphate ($PO_4$), and sulfide ($H_2S$) in coastal waters of Mimika, Indonesia due to human-based activities in this region. They have also observed a high salinity for seagrass and coral [11]. Abu El Ella et al. [12] collected ten surface water samples from different points in the southwest Giza area, Egypt. High concentrations of cadmium (Cd), copper (Cu), arsenic (As), and lead (Pb) were observed in

the collected water samples that exceeded the World Health Organization (WHO) drinking water guidelines [12]. They concluded that human and industrial activities played a significant role in polluting surface water bodies [12]. In other research, 17 water samples were collected from different points along India's coastline that extend about 7500 km [13]. The water quality index (WQI) results showed poor water quality in 13 of the 17 collected samples, which was mainly due to disposal of industrial waste, sewage, and agricultural runoff [13]. Qaroun Lake in Egypt is also suffering from uncontrolled release of industrial and domestic wastewater; high concentrations of aluminum (Al), cadmium (Cd), Zinc (Zn), and copper (Cu) have been reported for samples collected from the lake [14].

Most developing countries and countries in transition follow the pattern of supporting and enabling dramatic human-based activity growth (urbanization, industrial growth, agro-business, etc.) to overcome the rapid increase in population demand, with little/limited attention invested in pollution reduction or environmental protection interventions [15–18]. This leads to a dramatic deterioration in the environmental assets and natural resources, which creates the so-called "resource out of balance" scenario. This lack of balance between human-based activities and nature resource protection is clearly demonstrated in many coastal areas such as lakes, and has even expanded to become the most dominant problem at the global level and draw the attention of the United Nations (UN), with overcoming this problem now incorporated in the Sustainable Development Goals (SDGs) [19]. Many developments agencies have initiated programs to control and remediate the dramatic effects of such unbalanced growth of human activities and environmental assets' deterioration. For example, Horizon 2020 was initiated with a target of achieving a cleaner Mediterranean Sea by 2020 [20]. The Mediterranean Hotspots Investment Programme (MeHSIP) has also been initiated by the European Union with the purpose of managing waste in the Southern Mediterranean region, which will have a direct impact on de-polluting the Mediterranean Sea [21].

Lake Burullus is one of the lakes that suffers from such anthropogenic activities [22–24]. It is the second-largest lagoon out of five principal coastal lagoons in northern Egypt, with Lakes Maryut and Idku to its west and Lakes Bardawil and Manzala to its east [25,26]. The lake and its outskirt territories provide a habitat for a rich diversity of flora and fauna [27]. The main natural habitats that have been reported in the Burullus wetland include: salt marshes, hypersaline subkhas, sand formations, the lake proper, its islets, lake cuts, mud flats, reed beds, and swamps [28,29]. Due to its importance, it has been declared a protectorate and has been registered as a RAMSAR site (a wetland site selected to be of international importance under the RAMSAR Convention) [30]. Additionally, the lake has been recognized as an Important Bird Area by Birdlife International [31,32].

The size of the lake's aquatic surface area has reduced substantially over the years due to anthropogenic activities around the lake, such as agricultural reclamation and drying processes at the southern and eastern stretches [30,33,34]. In fact, the lake lost about 49% of its surface water area from 1801 (1092 km$^2$) to 1913 (556 km$^2$), and lost another 26% by 1997, reaching 410 km$^2$. In 2011, Dewidar indicated that the rate of loss of water body areas in the lake had been 2.7 km$^2$ per year for the period 1972–2007. However, the rate of loss had significantly decreased after 1998, when the lagoon was registered as a RAMSAR site [34].

The geomorphology of the area of the lagoon and its catchment basin is uniform and consists of a large deltaic plain intersected by natural and artificial watercourses [35]. As a result, the groundwater table throughout the region is very high, ranging from 0.5 to 1.5 m below ground level [36]. Moreover, groundwater in the northern half of the governorate is characterized by high salinity levels [37–39]. However, a decline in the salinity level was reported in Lake Burullus as a result of receiving a total quantity of 3904 million m$^3$/year agricultural drainage and industrial and domestic wastewater through nine inflows (eight drains and one canal named "Brimbal Canal") [40,41]. The decline in salinity level caused a reduction in the marine fish population (e.g., Liza ramada percentage by weight decreased from 16% to less than 1.8% from 1973 to 2003), increase in freshwater species (e.g., Tilapia percentage by weight increased from 81 to 98.2%), and

deterioration in the aquatic ecosystem [41,42]. High concentrations of total suspended solids (TSS) were reported in the lake, indicating high levels of pollution by non-organic and organic matter found in industrial and agricultural waste [41]. Lake Burullus also suffers from dense growth of aquatic plants, which is mainly due to nutrients' accumulation [43,44]. Another challenge is sediment accumulation around plant roots, which affects water circulation [41].

Many reports and research papers have indicated that Lake Burullus is suffering from pollution due to many anthropogenic activities around the lake (domestic, industrial, agriculture, fish farming, etc.). However, no work has been conducted to assess the contribution of each activity to the deterioration of the lake. This is mainly due to the interrelation of these activities, water reuse activities, the high number of stakeholders involved, and the inconsistency of measurements in terms of quality and quantity. Depolluting the impacts of such uncontrolled activities requires massive investment and it will take a prolonged period to restore the original and pristine environmental status. To develop a clear implementation strategy and investment plan to tackle such a complex situation, a clear assessment of the contribution of each type of activity is needed. The frequently asked question by developers and decision-makers that remains difficult to answer is how can we prioritize the necessary measures/intervention to reduce/mitigate the deterioration effect of interrelated human-based activities in order to achieve the target and countrywide vision? Budget limitations, the timescale, and available capacity considering other constraints make selecting the appropriate and most effective measure/intervention a complicated process. This study introduces an innovative/simplified integrated approach that links scientific and applied base approaches that facilitate the identification, quantification, and prioritization of the polluter within interrelated human-based activities. The weighted contributions of agriculture, urban, industrial, and fish farming sectors are assessed in terms of chemical oxygen demand (COD), total nitrogen (TN), and total phosphorus (TP). The proposed approach not only aids feasibility and investment planning but also allows for monitoring and evaluation stages during and after measures/interventions' implementation, for continuous improvement.

## 2. Materials and Methods

### 2.1. Study Area

Lake Burullus is located in the north of the Nile Delta in the Kafr El Sheikh governorate, Egypt [45]. It is bordered by the Mediterranean coast from the north, agricultural lands from the south, the Rosetta branch of the River Nile from the west, and the Damietta branch of the River Nile from the east [5]. The protectorate has a total area of 460 km$^2$ (as shown in Figure 1) [45], defined by the following coordinates: 31°36′ N 31°07′ E in the northeast, 31°22′ N 30°33′ E in the southwest, 31°36′ N 30°33′ E in the northwest, and 31°22′ N 31°07′ E in the southeast [29]. The width of the lake from south to north varies from one side to the other. The western part has the least width, which does not exceed 5 km; then, the width increases in the middle sector to reach an average of 11 km, while the maximum width of the lake is about 16 km [28,46]. The area of the lake is about 410 km$^2$, while the remaining area represents the islets, which total about 50 km$^2$, and a belt of sand dunes (separates the lake shore in the south from the Mediterranean coast in the north) that has a width ranging from 0.5 to 1 km and a shoreline of about 65 km long [31].

The depth of the lake varies from 40 cm near the shores to 200 cm near the Al Boughaz outlet, which connects the lake to the Mediterranean Sea [41]. The lake currently holds an average water volume of about 328 million m$^3$ at zero mean sea level [33,34].

### 2.2. Identification of the Main Human-Based Activities

The most important human-based activities that dump into the network of drains leading to Lake Burullus are drainage wastewater, domestic wastewater, industrial wastewater, fish farm waste, and solid waste [47].

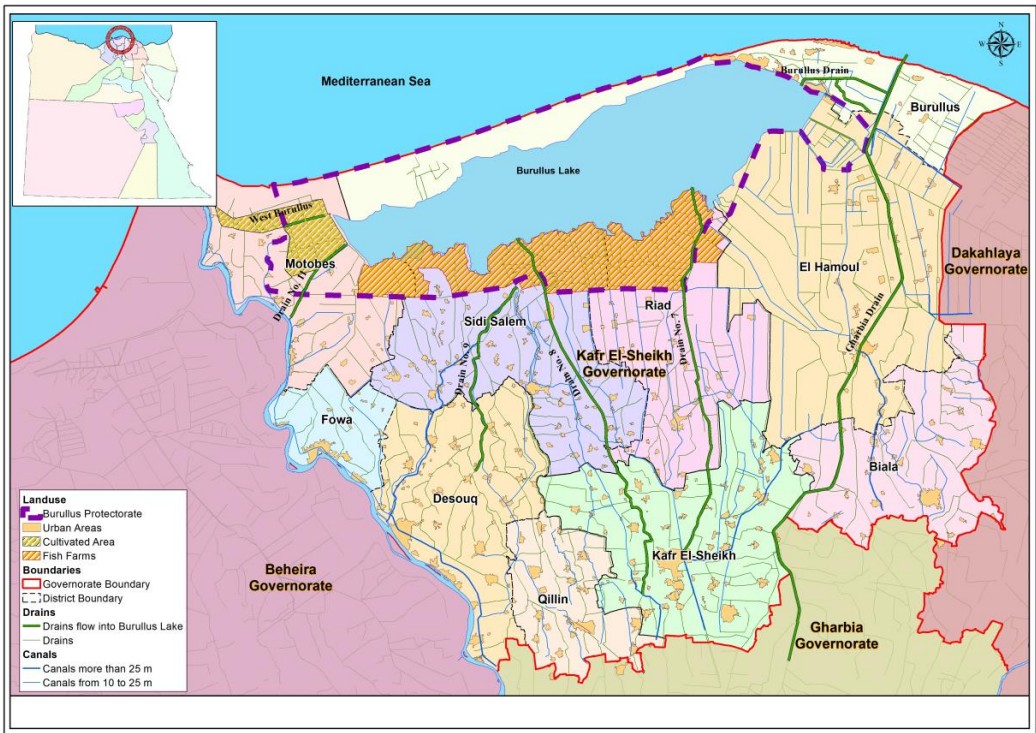

**Figure 1.** Lake Burullus Protectorate and seven main agricultural drains discharging to the lake.

### 2.3. Data Collection

Data and reports on activities and pollutants were collected from the following relevant stakeholders: Ministry of Irrigation and Water Resources (MIWR), Egyptian Environmental Affairs Agency (EEAA), Ministry of Agriculture and Land Reclamation (MALR), affiliated (subsidiary) companies of the Holding Company for Water and Wastewater, and users of the agricultural drains (beneficiaries–polluters).

### 2.4. Determination of the Catchment Areas

The catchment areas of the main and lateral drains to Lake Burullus were determined in order to establish each investigated activity's area of influence (zone) on these catchment areas.

### 2.5. Determination of the Water Flow in the Drainage Network

The water flow in the drainage network (main and lateral drains) was determined for the following activities:

- Industrial activities: by identifying the locations of industrial facilities and the amount of industrial wastewater discharged to the drainage system.
- Agricultural activities: by calculating the amount of water required for irrigation of cultivated lands within the catchment area of the drainage network and estimating the irrigation water losses through the drainage system.
- Domestic activities: by identifying the locations of wastewater treatment plants (WWTPs) in the drainage network and calculating the daily amount of treated wastewater pumped to the drainage system.
- Fish farming activities: these activities were not included as generating an input flow to the drainage system because they depend on water from the drainage network for fish feeding (as stipulated by Law 124/1983); they instead act as internal recirculation to the drainage system.

### 2.6. Selection of the Main Pollutants' Indicators

Chemical oxygen demand (COD), total nitrogen, and total phosphorus were selected as being the most important parameters to indicate the deterioration of the water quality in the drains and lake [48,49]. High concentrations of these parameters in waterways reduce the oxygen and cause overgrowth of algae and water plants in a way that leads to disturbance of the ecological balance of the aquatic system [50,51]. Biological elements (e.g., pathogens) were excluded from the study due to the difficulty in obtaining regular data on them and due to their dynamic growth within the ecological system (when there are favorable growth conditions). The COD indicator was considered sufficient in this regard as it implies the impact of pathogens in the case of untreated wastewater.

### 2.7. Estimation of the Pollutants Load Generated by Each Activity

#### 2.7.1. Industrial Activities

We calculated the pollutant loads from measured concentrations at the outlets of the industrial facilities; these loads were discharged directly into the drainage network in the study area.

#### 2.7.2. Domestic Activities

Direct and indirect pollutant loads are normally generated from domestic activities in the study region:

- Direct load: originates from wastewater treatment plants (WWTPs), with the worst scenario when the produced wastewater is discharged to the drainage network completely untreated (COD: 300 mg/L), and the best scenario when these plants operate at full efficiency to meet the required effluent standards (COD effluent: 80 mg/L). Equation (1) is used for COD load calculation.

$$\text{COD load in kg/day} = (\text{BOD in mg/L} \times \text{effluent volume in m}^3/\text{day})/1000 \quad (1)$$

- Indirect load: loads resulting from the unserved population within the drainage network area, which can be calculated using published per capita generation rates for COD, TN, and TP in Egypt [52]. The sewage generated in this case is disposed of to the drainage system either through negative discharging or via vacuum trucks.

#### 2.7.3. Fish Farming Activities

The pollutant loads generated from fish farms were estimated based on the commune operational practice of fish farming, including fish feeding, water feeding, and water discharging (water cycle). The pollutant load was converted per feddan and used as a unit to calculate the total pollutant loads generated from the fish farms around Lake Burullus. These values depend mainly on the cultivated types of fish, fodder, fish density, and cultivation conditions.

#### 2.7.4. Solid Wastes

It was not possible to calculate the pollution loads of COD, TN, and TP resulting from the disposal of solid wastes into the drainage network, and consequently, the lake due to the difficulty of estimating their values from the available data.

#### 2.7.5. Agricultural Activities

Calculation of TN and TP loads was based on the fertilizers (N, P) quantities required for crop production within the limits of the drainage system and estimation of the losses of these fertilizers into the drainage network. TN and TP losses through soil leaching were calculated by using the regression model developed by De Willigen [53], as shown in Equations (2) and (3).

$$\text{TN}_{\text{leaching}} \ (\text{kg N/ha/year}) = (0.0463 + 0.0037 \times (\frac{P}{C \times L})) \times (N_f + \acute{\alpha} \times N_{om} - N_u) \quad (2)$$

where P is the annual precipitation (mm/year), C the clay content (%), L the layer thickness or rooting depth (m), $N_f$ the mineral and manure TN (kg N or P/ha), $\acute{a}$ the decomposition rate, Nom the amount of TN in soil organic matter (kg N/ha), and Nu the TN uptake by harvested crop and crop residues removed from the field (kg N/ha per year).

$$TP_{leaching} \ (kg \ P/ha/year) = (0.0463 + 0.0037 \ \times \ (\frac{P}{C \ \times \ L})) \ \times \ (P_f + \acute{a} \ \times \ P_{om} - P_u) \qquad (3)$$

where $P_f$ is the mineral and manure TP (kg P/ha), $P_{om}$ the amount of TP in soil organic matter (kg P/ha), and $P_u$ the TP uptake by harvested crop and crop residues removed from the field (kg P/ha per year).

The following values were used for each variable shown in Equations (2) and (3):

1.  An annual precipitation of 100 mm was used in subsequent calculations [54]. An average clay content of 11% in the delta region was used in subsequent calculations [55]. The rooting depth for each crop was obtained from different sources, as detailed in Supplementary Table S1.
2.  The amounts of nutrients in the soil organic matter were decided as follows:

    *   Nitrogen in soil organic matter was calculated by using Equation (4):

$$N_f = \rho \times L \times TN_{in \ soil} \qquad (4)$$

    where $\rho$ is the bulk density (kg/m$^3$) of the soil, and TN is the total nitrogen in the soil (kg/kg).

    *   Phosphorus in soil organic matter was calculated by using Equation (5):

$$P_f = \rho \times L \times TP_{in \ soil} \qquad (5)$$

    where TP is the total phosphorus in the soil (kg/kg). The bulk density used in subsequent calculations was 1790 kg/m$^3$ [56], while the total TN and total TP in the soil were 0.000125 kg N/kg and 0.000142 kg P/kg, respectively [57,58].

3.  The decomposition rate in equations (2) and (3) for nitrogen was assumed to be 1.6% per year [53], and for phosphorus, was assumed to be 1.9% per year [59].

    Gruhn et al. (2000) stated that the TN uptake efficiency by crops was less than 30% in the developed world and 20% in developing countries. Based on this literature, the TN uptake efficiency in developing countries is expected to reach 20% in 2025 [60]. Consequently, it is assumed that the TN uptake by different crops will reach 15%. Beegle and Durst (2002), meanwhile, stated that the efficiency of plant uptake in phosphorus is very low, even with optimum management, usually less than 20%; hence, the TP uptake by the different crop species is assumed to be 15% [61].

4.  The mineral and manure contents in nutrients (TN, TP), as shown in Equations (2) and (3) above, were calculated on the basis of the following considerations:

    *   The total fertilizers used per unit area of crops was obtained from a Food and Agriculture Organization (FAO) report from 2005 [62]. Based on this report, fertilizer consumption increased by about 60% in 24 years from 1980 to 2004. From 2004 till the present, fertilizer consumption is expected to have been increased by about 30%. Hence, the fertilizers used in subsequent calculations were multiplied by a factor of 1.3 to account for the increase in consumption of nitrogen and phosphate fertilizers. Tolba and Saab (2008) also support our assumption when stating that Egypt uses the highest rate of fertilizers (more than 900 kg of fertilizers per hectare per year) compared to other Arab countries [63]. This is mainly due to the introduction of new high-yielding varieties of crops, which need higher rates of fertilizer utilization. The annual quantities of fertilizers used per unit area for various crops are listed in Supplementary Table S2.
    *   The recommended amounts of farmyard manure (FYM, in t/ha) required to cultivate the prevailing crops were obtained from different references. Prakash and Singh (2013) stated that well-decomposed FYM may contain nitrogen from 1.2 to 2.0% and phosphorus from 0.5 to 0.7% [64]. The amounts of nitrogen and phosphorus applied to each crop from FYM were calculated by multiplying the amount of FYM by 0.02 and 0.007, respectively. The annual quantity of FYM used per unit area of each crop is listed in Supplementary Table S3. The amounts of nitrogen and phosphorus in FYM application were calculated by using Equations (6) and (7).

Amount of TN contained in FYM (t/ha) = Total amount of FYM applied (t/ha) $\times$ 0.02 $\qquad$ (6)

$$\text{Amount of TP contained in FYM (t/ha)} = \text{Total amount of FYM applied (t/ha)} \times 0.007 \qquad (7)$$

*2.8. Pollutant Loads' Verification*

We compared the calculated/estimated pollutant loads from the main human-based activities with the measured pollutant loads at the tail-ends of the agricultural drains (completely mixed pollutant loads from different activities) for verification. We analyzed and comparing the generated pollutant loads from the different human-based activities and identified the activity that generates the highest pollutant values for COD, TN, and TP.

**3. Results and Discussion**

*3.1. Assessment of Drainage Water Flow into Lake Burullus*

Lake Burullus serves as a reservoir for drainage water. The lake receives drainage water from agricultural lands south of the lake through the seven main drains depicted in Figure 1 (i.e., West Burullus drain, Drain 11, Drain 9, Drain 8, Drain 7, Terah drain, and Burullus drain) in addition to fresh water from the Bramble Canal situated at the western part of the lake. According to the Egyptian Environmental Affairs Agency and the Ministry of Water Resources and Irrigation, the Burullus drain receives about 15% of the Gharbia drain flow, while the rest of this flow discharges directly to the Mediterranean Sea. Moreover, Burg El Burullus is the only WWTP that discharges directly to the lake.

Table 1 illustrates the drainage catchment areas for the agricultural drains in the Nile Delta discharging to Lake Burullus, including the crop types and areas corresponding to each one of these drains. The agricultural water inflows were calculated in this study on the basis of the amounts of standard water demand for the crops in the cultivated lands within the catchment area of the drainage network and a subsequent estimation of the resulting drainage water, i.e., the irrigated water lost to the drainage system (30% used to estimate the drainage water). The agricultural water inflows in the summer and winter seasons were calculated as 0.91 and 0.37 billion m$^3$/year, respectively, with a total flow of about 1.27 billion m$^3$/year. According to Table 1, the maximum rate of water discharged to the lake is during the rice cultivation season (July–September), while the minimum rate is in January and February. Moreover, Drain 9 is shown to discharge the maximum amount of drainage water, while the Burullus drain discharges the minimum. There are 16 industrial establishments discharging wastewater with a total flow of about 4.5 million m$^3$/year to the network of agricultural drains in the catchment basin of Lake Burullus. There are 24 WWTPs discharging their effluents to the catchment basin of Lake Burullus, as well as the rural population that is not served with sewered sanitation services. The total amount of domestic wastewater discharged to the drainage system is about 50 million m$^3$/year. Consequently, the calculated total inflow to the lake from the three activities (agricultural, domestic, and industrial) is about 1.3 billion m$^3$/year. These results are summarized in the last row of Table 2. The first two rows of Table 2 present a range of reported values for the inflows of drainage water to the seven main drains discharging to Lake Burullus (3.2–4.7 billion m$^3$/year). These values reflect actual measurements at the pump stations, which are located at the tail-ends of the agricultural drains that transport drainage water to Lake Burullus.

**Table 1.** Catchment area of each agricultural drain in feddan with the corresponding crops.

| | Item | Water Requirement (m³/feddan) | Terah | | Drain 7 | | Drain 8 | | Drain 9 | | Drain 11 | | Gharbia | |
|---|---|---|---|---|---|---|---|---|---|---|---|---|---|---|
| | | | Crop Area, Feddan | Flow, m³/year | Crop Area, Feddan | Flow, m³/year | Crop Area, Feddan | Flow, m³/year | Crop Area, Feddan | Flow, m³/year | Crop Area, Feddan | Flow, m³/year | Crop Area, Feddan | Flow, m³/year |
| Summer crops | Maize | 3507 | 944 | 3.31 | 4537 | 15.91 | 14,271 | 50.05 | 6629 | 23.25 | 5123 | 17.97 | 35,305 | 123.82 |
| | Sorghum | 3544 | 0 | 0 | 0 | 0.000 | 0 | 0.000 | 0 | 0.000 | 0 | 0.000 | 0 | 0.001 |
| | Rice | 9292 | 37,797 | 351.2 | 43,416 | 403.42 | 78,578 | 730.14 | 44,327 | 411.89 | 37,097 | 344.70 | 175,697 | 1632.58 |
| | Yellow Maize | 3507 | 0 | 0 | 99 | 0.346 | 504 | 1.766 | 67 | 0.236 | 270 | 0.947 | 4874 | 17.095 |
| | Peanut | 4697 | 0 | 0 | 0 | 0.000 | 0 | 0.000 | 0 | 0.000 | 0 | 0.000 | 577 | 2.710 |
| | Sesame | 3014 | 0 | 0 | 0 | 0.000 | 4 | 0.011 | 4 | 0.011 | 22 | 0.067 | 4 | 0.012 |
| | Soybean | 3344 | 0 | 0 | 3 | 0.010 | 11 | 0.037 | 7 | 0.025 | 10 | 0.034 | 33 | 0.111 |
| | Sunflower | 2622 | 0 | 0 | 0 | 0.000 | 5 | 0.013 | 0 | 0.000 | 0 | 0.000 | 1 | 0.002 |
| | Onion—summer | 4497 | 0 | 0 | 0 | 0.000 | 0 | 0.000 | 0 | 0.000 | 0 | 0.000 | 13 | 0.059 |
| | Potato | 2600 | 1 | 0.003 | 15 | 0.040 | 6044 | 15.715 | 61 | 0.159 | 28 | 0.073 | 8790 | 22.855 |
| | Tomato | 4155 | 178 | 0.741 | 309 | 1.284 | 1026 | 4.263 | 412 | 1.712 | 531 | 2.205 | 1199 | 4.982 |
| | Other vegetables | 3260 | 784 | 2.555 | 5108 | 16.652 | 14,489 | 47.236 | 8576 | 27.957 | 12,972 | 42.290 | 18,784 | 61.235 |
| | Other crops | 3099 | 95 | 0.294 | 89 | 0.275 | 748 | 2.319 | 176 | 0.546 | 317 | 0.983 | 298 | 0.924 |
| | Cotton | 3977 | 8152 | 32.42 | 7561 | 30.07 | 16,314 | 64.88 | 13,078 | 52.01 | 6169 | 24.53 | 27,731 | 110.28 |
| Total flow (summer), million m³/year | | | | 390.53 | | 468.01 | | 916.43 | | 517.79 | | 433.80 | | 1976.66 |
| Winter crops | Long clover | 3407 | 9597 | 32.69 | 12,209 | 41.596 | 43,982 | 149.848 | 13,262 | 45.183 | 19,266 | 65.640 | 78,953 | 268.994 |
| | Wheat | 2034 | 19,307 | 39.27 | 32,321 | 65.742 | 82,705 | 168.223 | 44,059 | 89.615 | 38,729 | 78.774 | 149,515 | 304.114 |
| | Barley | 1752 | 33 | 0.059 | 307 | 0.538 | 224 | 0.392 | 19 | 0.033 | 855 | 1.498 | 233 | 0.409 |
| | Broad bean | 1583 | 6975 | 11.04 | 3421 | 5.416 | 7895 | 12.498 | 3538 | 5.601 | 4594 | 7.273 | 28,118 | 44.510 |
| | Lentil | 1779 | 0 | 0 | 0 | 0 | 0 | 0 | 0 | 0 | 0 | 0 | 9 | 0.016 |
| | Fenugreek | 1583 | 0 | 0 | 0 | 0 | 0 | 0 | 0 | 0 | 0 | 0 | 0 | 0 |
| | Chickpea | 1779 | 0 | 0 | 0 | 0 | 0 | 0 | 0 | 0 | 0 | 0 | 0 | 0 |

**Table 1.** *Cont.*

| Item | Water Requirement (m³/feddan) | Terah | | Drain 7 | | Drain 8 | | Drain 9 | | Drain 11 | | Gharbia | |
|---|---|---|---|---|---|---|---|---|---|---|---|---|---|
| | | Crop Area, Feddan | Flow, m³/year | Crop Area, Feddan | Flow, m³/year | Crop Area, Feddan | Flow, m³/year | Crop Area, Feddan | Flow, m³/year | Crop Area, Feddan | Flow, m³/year | Crop Area, Feddan | Flow, m³/year |
| Lupin | 1779 | 0 | 0 | 0 | 0 | 0 | 0 | 0 | 0 | 0 | 0 | 0 | 0 |
| Flax | 1620 | 202 | 0.327 | 318 | 0.515 | 327 | 0.530 | 130 | 0.210 | 47 | 0.076 | 2715 | 4.39 |
| Winter onion | 2216 | 3 | 0.007 | 1 | 0.003 | 14,360 | 31.82 | 54 | 0.120 | 3 | 0.006 | 8980 | 19.90 |
| Garlic | 1615 | 0 | 0 | 0 | 0 | 70 | 0.113 | 0 | 0 | 0 | 0 | 15 | 0.024 |
| Sugar beet | 2626 | 16,121 | 42.34 | 15,189 | 39.89 | 13,550 | 35.58 | 8445 | 22.178 | 5319 | 13.967 | 33,214 | 87.22 |
| Potato | 2212 | 32 | 0.071 | 56 | 0.123 | 2928 | 6.48 | 120 | 0.266 | 106 | 0.235 | 5910 | 13.07 |
| Tomato | 2043 | 155 | 0.316 | 39 | 0.080 | 31 | 0.063 | 20 | 0.040 | 427 | 0.872 | 235 | 0.480 |
| Other vegetables | 2699 | 412 | 1.113 | 320 | 0.863 | 3719 | 10.039 | 500 | 1.351 | 627 | 1.692 | 4404 | 11.89 |
| Tahrish | 3030 | 6430 | 19.48 | 4918 | 14.90 | 8852 | 26.82 | 4503 | 13.64 | 2207 | 6.68 | 15,111 | 45.79 |
| Other crops | 2699 | 4 | 0.011 | 1 | 0.003 | 861 | 2.32 | 2 | 0.005 | 0 | 0.001 | 701 | 1.89 |
| Total flow (winter), million m³/year | | | 146.73 | | 169.66 | | 444.73 | | 178.25 | | 176.72 | | 802.71 |
| Total flow (summer and winter), million m³/year | | | 537.26 | | 637.68 | | 1361.16 | | 696.04 | | 610.52 | | 2779.37 |
| 30% of the flow reaching Lake Burullus, million m³/year | | | 161.18 | | 191.30 | | 408.35 | | 208.81 | | 183.16 | | 125.01 * |
| Total flow to Lake Burullus, million m³/year | | | | | | 1277.87 | | | | | | | |

* 15% of Al Gharbia drain flow discharge to Lake Burullus.

**Table 2.** Inflows from different activities to drains discharging to Lake Burullus (million m$^3$/year).

| No. | Flow, Million m$^3$/year | Drains | | | | | | | | Total to LB | Reference |
|---|---|---|---|---|---|---|---|---|---|---|---|
| | | Terah | Burullus 15% Gharbia | Drain 7 | Drain 8 | Drain 9 | Drain 11 | West Burullus | Lake Burullus * | | |
| 1 | Q total | 672.0 | 68.0 | 502.9 | 391.6 | 785.6 | 634.7 | 176.8 | | 3231.7 | [65] |
| 2 | Q total | | | 690.3 | 542.4 | 1934.4 | 1419.1 | 138.0 | | 4724.2 | [66] |
| 3 | Q agriculture summer | 117.2 | 88.9 | 140.4 | 274.9 | 155.3 | 130.1 | | | 906.9 | This study |
| | Q agriculture winter | 44.0 | 36.1 | 50.9 | 133.4 | 53.5 | 53.0 | | | 371.0 | |
| | Q agriculture total | 161.2 | 125.1 | 191.3 | 408.3 | 208.8 | 183.2 | | | 1277.9 | |
| | Q domestic | 0.7 | 8.9 | 15.3 | | 21.2 | 2.2 | | 1.8 | 50.1 | |
| | Q industrial | | 3.8 | 0.1 | | 0.6 | | | | 4.5 | |
| | Q total | 161.9 | 137.8 | 206.7 | 408.3 | 230.6 | 185.4 | | 1.8 | 1332.5 | |

* Burg El Burullus WWTP, which discharges its effluents directly to Lake Burullus.

The difference between the amount of drainage water when assessed on the basis of crop water demands for flow into the drainage network within the project area, and the amount(s) of drainage water measured through the pumping stations at the tail-ends of the agricultural drains discharging to Lake Burullus, may be attributed to:

1.  The use of water exceeding the actual needs for irrigating crops; this was reported in some field studies in the project area [36].
2.  The use of water from irrigation canals for fish farming in some areas instead of drainage water (as per Law 124/1983 for fish farming), which is consequently added to the amount of water flowing through the drainage network (the fish farming area amounts to 25,000 feddans (105 km$^2$), where the water is replaced once every 15 days at 1 m depth; this cycle contributes 2 billion m$^3$/year of additional wastewater to the drainage system).
3.  A possible underestimation in the figure of 30% used to estimate the drainage water, i.e., the irrigated water lost to the drainage system [36].
4.  A change in the types of crops, and hence, irrigation water requirements, pertinent to new crops that might violate the agreed allocation of crops to each governorate. This change may have taken place because of higher revenues for some crops (e.g., rice).
5.  A lack of continuous measurements at pumping stations at the tail-ends of the drains (e.g., to record changing flows in different years according to the water inflow from the Nile River).
6.  A lack of measured flows at the lateral (branch) drains, and also, at mixing points of the drainage water on the main irrigation canals.
7.  Inaccurate data on rice acreage distributed over the administrative districts of the Kafr El Sheikh, Gharbia, and Dakahlia governorates and officially registered by the agriculture departments of the Ministry of Agriculture and Land Reclamation (MALR). The actual rice acreage is most likely much higher (up to three times) than the officially registered area. Accordingly, the drainage wastewater estimated in this study on the basis of the cropping area over the drainage catchment area is most likely much less (up to three times). As the Kafr El Sheikh governorate had the largest rice cultivated area, which is considered to be the most water-consuming crop, the Ministry of Water Resources and Irrigation (MWRI) in coordination with the MALR have implemented a policy to reduce the rice-cultivated areas, and also, replace rice crops with other less water-consuming crops. Moreover, the MALR is currently checking for inconsistencies concerning farmers who have violated this policy.

There are many contradictory data related to the flow of drainage due to high seasonal variation and very dynamic flow characteristics; therefore, the most realistic figures were selected in this study based on actual agricultural areas that consume the water.

### 3.2. Pollution Components

### 3.2.1. Industrial Wastewater Pollution Component

The assessment of pollution in Lake Burullus due to industrial wastewater inflows was based on data from periodic monitoring and audit activities carried out regularly by the Egyptian Environmental Affairs Agency (EEAA) on all industrial establishments discharging their industrial effluents to the network of agricultural drains leading to the lake. There are 16 industrial establishments discharging their industrial wastewater to the network of agricultural drains in the catchment basin of Lake Burullus. Most of them have pretreatment facilities to treat the industrial wastewater, in compliance with the requirements of Law 48/1982, before they discharge it to the nearest agricultural drain; the rest are non-compliant with these requirements, only working toward implementing the necessary pretreatment. Table 3 presents EEAA data on sampled COD concentrations, industrial wastewater outflows, and pollution loads from the 16 industrial establishments discharging their industrial wastewater indirectly to Lake Burullus through the network of agricultural drains that lead to the lake. In total, 14 industrial establishments discharge to the Al Gharbia drain, which next discharges 15% of its flow to Lake Burullus via the Burullus Drain. Alongside this, two industrial establishments, located within the Kafr El Sheikh and Gharbia governorates, discharge to Drains 7 and 9, respectively. The total industrial wastewater flow is 4.44 million m$^3$/year, contributing an annual COD pollution load of 1163.4 tons.

**Table 3.** Flows and pollution loads from the 16 industrial establishments discharging to Lake Burullus.

| No. | Drain | Governorate | Company | Flow (m³/day) | Flow (mm³/year) | COD Conc. (mg/L) | Total Effluent COD Load (ton/year) | Permissible COD Load (ton/year) | Required COD Load to Be Removed (ton/year) |
|---|---|---|---|---|---|---|---|---|---|
| 1 | | | Oil and soap company | 300 | 0.11 | 76 | 8.3 | 8.8 | 0.0 |
| 2 | | | Electronic industries | 150 | 0.05 | 160 | 8.8 | 4.4 | 4.4 |
| 3 | | | Paperboard company | 30 | 0.01 | 520 | 5.7 | 0.9 | 4.8 |
| 4 | | | Mill company | 50 | 0.02 | | | | |
| 5 | | | Spinning and weaving company | 60,000 | 21.90 | 333.3 | 7299.3 | 1752.0 | 5547.3 |
| 6 | | Al Gharbia | Spinning and weaving company | 4000 | 1.46 | 89 | 129.9 | 116.8 | 13.1 |
| 7 | Al Gharbia drain | | Textiles company | 300 | 0.11 | 180 | 19.7 | 8.8 | 11.0 |
| 8 | | | Agro industry | 80 | 0.03 | 155 | 4.5 | 2.3 | 2.2 |
| 9 | | | Dyeing & finishing company | 350 | 0.13 | 296 | 37.8 | 10.2 | 27.6 |
| 10 | | | Dyeing company | 200 | 0.07 | | | | |
| 11 | | | Textiles company | 150 | 0.05 | 784 | 42.9 | 4.4 | 38.5 |
| 12 | | | Textiles company | 160 | 0.06 | 480 | 28.0 | 4.7 | 23.4 |
| 13 | | | Textiles company | 180 | 0.07 | 163 | 10.7 | 5.3 | 5.5 |
| 14 | | Kafr El Sheikh | Sugar company | 9600 | 1.15 | 64 | 73.7 | 92.2 | |

**Table 3.** *Cont.*

| No. | Drain | Governorate | Company | Flow (m³/day) | Flow (mm³/year) | COD Conc. (mg/L) | Total Effluent COD Load (ton/year) | Permissible COD Load (ton/year) | Required COD Load to Be Removed (ton/year) |
|---|---|---|---|---|---|---|---|---|---|
| | Total to Al Gharbia drain | | | 75,550 | 25.2 | | 7669.4 | | 5677.7 |
| 15 | Drain 7 | Kafr El Sheikh | Oil and soap company | 301.2 | 0.11 | 60 | 2.2 | | |
| 16 | Drain 9 | Al Gharbia | Oil and soap company | 1506 | 0.55 | 60 | 10.8 | | |
| Total to Lake Burullus (only 15% of Al Gharbia drain to LB) | | | | 13,139.7 | 4.5 | | 1163.4 | | 851.6 |

### 3.2.2. Agricultural Wastewater Pollution Component

The present study determined the quantities of TN and TP flowing in the agricultural drains leading to Lake Burullus through (a) estimation of the quantities of crops cultivated within the catchment areas of these drains, using the statistics of the Ministry of Agriculture and Land Reclamation, (b) assessment of the amounts of fertilizers required to produce the above quantities of crops, and (c) evaluation of the proportion of these fertilizers lost in agricultural drainage water. Table 4 presents the TN and TP loads generated from the agricultural areas corresponding to each crop. The highest TN load of 5222 tons/year was generated from summer crops, while the highest TP load of 1917 tons/year was generated from winter crops. Agricultural areas cultivated by rice represent about 73% and 60% of the TN and TP loads generated in the summer season, respectively, while agricultural areas cultivated by wheat and long clover combined represent about 79% and 65% of the TN and TP loads generated in the winter season, respectively. The annual quantities of TN and TP generated from all agricultural areas and eventually discharged to Lake Burullus were assessed to be about 9782 and 3659 tons/year, respectively.

### 3.2.3. Domestic Wastewater Pollution Component

Agricultural drains do not carry only agricultural drainage water to Lake Burullus. According to the Egypt National Rural Sanitation Strategy of 2008, WWTPs discharge their treated effluents to the nearest agricultural drain [67]. Pollution due to domestic wastewater is considered to emanate from two types of discharge to the agricultural drains leading to the lake as follows:

- Direct discharge: This is the discharge to the nearest agricultural drain of treated effluents from WWTPs serving cities and villages located within the catchment areas of the receiving drains. Table 5 lists 24 such WWTPs, their governorates, as well as the quantity and receptor (final disposal outlet) of their treated effluent. The total flow of effluents from WWTPs is about 50 million m³/year. The organic COD loads from the WWTPs were estimated considering their actual effluent flow in Table 5 and assuming that the best scenario is when the preceding plants operate at full efficiency (COD effluent: 80 mg/L) to meet the required effluent standards (Law 48/1982) [68], while the worst scenario is when the generated wastewater is discharged to the drainage network completely untreated (COD: 300 mg/L). The loads of TN and TP were calculated on the basis of the values TN = 40 mg/L, TP = 5 mg/L, considering that the current effluent standards (Law 48/1982) do not require these WWTPs to treat and dispose of such elements. A law was proposed to introduce lower maximum values for TN and TP in the WWTP effluents (TN = 10 mg/L, TP = 2 mg/L), but it did not pass due to a lack of funds to meet the upgrade cost of existing WWTPs and the increased cost of new WWTPs.
- Indirect discharge: This is the discharge to agricultural drains of untreated domestic wastewater from a rural population that is not served with sewered sanitation services within the catchment areas of the receiving drains. There are two ways in which such untreated domestic wastewater usually ends up in agricultural drains leading to Lake Burullus: (1) pipe networks to lower the groundwater table (negative discharge), and

(2) vacuum trucks. The COD, TN, and TP loads of the indirect discharge polluting Lake Burullus were assessed by using the average COD, TN, and TP generation rates shown in Figure 2. Furthermore, we assumed that only half of these pollutants would, eventually, reach the agricultural drains discharging to the lake. This assumption was based on the following considerations: (a) anaerobic degradation of sewage in vaults/septic tanks, (b) groundwater seepage into vaults/septic tanks, (c) losses due to street flooding, and (d) losses during evacuation and cleaning.

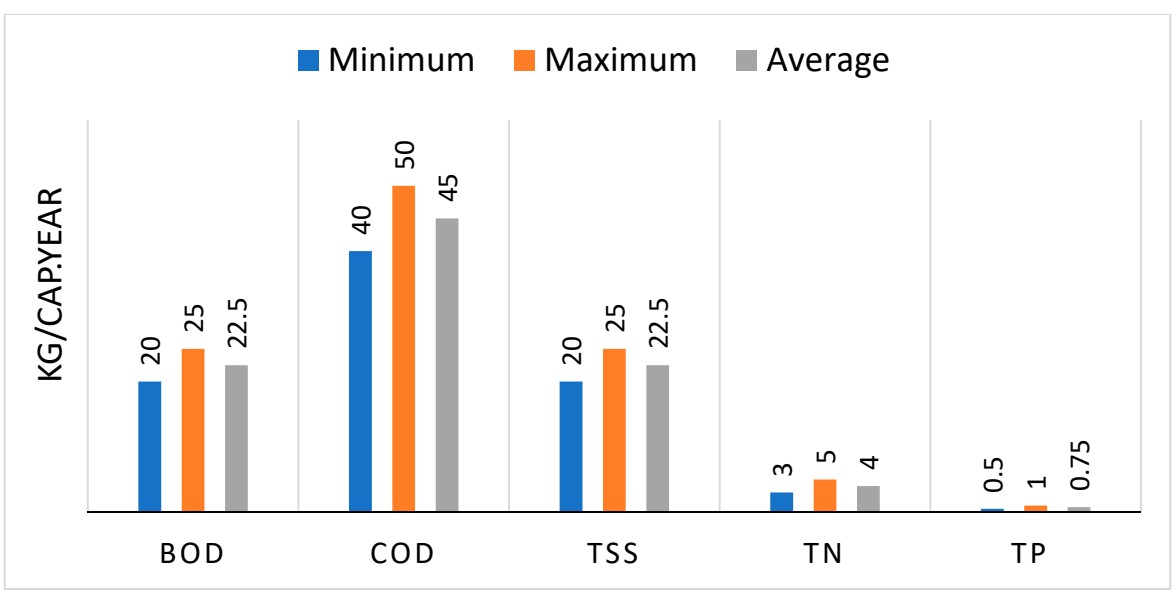

**Figure 2.** Generation rates for BOD, COD, TSS, TN, and TP pollutants in Egypt [52].

Table 6 includes the results of our domestic wastewater pollution assessment. The total COD load to Lake Burullus from both direct and indirect discharge ranges between 50,922 tons/year for the best scenario and 61,930 tons/year for the worst scenario. The results also show that the TN pollution load is about 6172 tons/year, while the TP pollution load is about 1032 tons/year.

### 3.2.4. Fish Farm Pollution Component

Law no. 124/1983 on the organization of fish farms and their implementing regulations, as well as Ministerial Decree no. 329/1985, identified areas for fish farming in the northern lakes of Egypt (Lake Manzala, Lake Burullus, Lake Edco, and Lake Mariut). This legal framework provided the main driving force behind the boost of fish-farming activity in Egypt as its main objective was to close a large proportion of the food (protein) gap in the country. Since then, aquaculture has grown to reach 70% of the total fish production (919,585 tons/year) as compared to only 30% from natural fisheries (385,209 tons/year), and has led to an annual share of 16.44 Kg/capita from the available domestic fish production [69].

**Table 4.** TN and TP loads generated from agricultural areas for each crop.

| Summer Crop | Terah | Drain 7 | Drain 8 | Drain 9 | Drain 11 | Gharbia | Total Area, Feddan | Nf, kg/ha (Fertilizers + FYM) | Pf, kg/ha (Fertilizers + FYM) | L, m | Nu | Pu | TN Total, ton | TP Total, ton |
|---|---|---|---|---|---|---|---|---|---|---|---|---|---|---|
| Maize | 944 | 4537 | 14,271 | 6629 | 5123 | 5296 | 36,800.6 | 377 | 72 | 0.4 | 94.3 | 21.5 | 570.1 | 100.9 |
| Sorghum | 0 | 0 | 0 | 0 | 0 | 0.24 | 0.035709 | 416 | 91 | 0.2 | 104.1 | 27.4 | 0.001 | 0.0002 |
| Rice | 37,797 | 43,416 | 78,578 | 44,327 | 37,097 | 26,355 | 267,568.9 | 189 | 52 | 0.2 | 47.2 | 15.6 | 3412.0 | 878.4 |
| Yellow maize | 0 | 99 | 504 | 67 | 270 | 731 | 1670.9 | 377 | 72 | 0.4 | 94.3 | 21.5 | 25.9 | 4.6 |
| Peanut | 0 | 0 | 0 | 0 | 0 | 87 | 86.5 | 124 | 91 | 0.5 | 31.0 | 27.4 | 0.38 | 0.26 |
| Sesame | 0 | 0 | 4 | 4 | 22 | 1 | 30.4 | 46 | 10 | 0.5 | 11.6 | 3.0 | 0.05 | 0.01 |
| Soybean | 0 | 3 | 11 | 7 | 10 | 5 | 36.9 | 124 | 91 | 0.5 | 31.0 | 27.4 | 0.16 | 0.11 |
| Sunflower | 0 | 0 | 5 | 0 | 0 | 0 | 5.08 | 416 | 91 | 0.8 | 104.1 | 27.4 | 0.059 | 0.012 |
| Onion—summer | 0 | 0 | 0 | 0 | 0 | 2 | 1.95 | 189 | 52 | 0.3 | 47.2 | 15.6 | 0.018 | 0.005 |
| Potato | 1 | 15 | 6044 | 61 | 28 | 1319 | 7468.5 | 390 | 189 | 0.5 | 97.6 | 56.7 | 104.3 | 47.0 |
| Tomato | 178 | 309 | 1026 | 412 | 531 | 180 | 2635.8 | 390 | 143 | 0.5 | 97.6 | 43.0 | 36.8 | 12.6 |
| Other vegetables | 784 | 5108 | 14,489 | 8576 | 12,972 | 2818 | 44,746.5 | 325 | 280 | 0.3 | 81.4 | 84.0 | 726.7 | 582.6 |
| Other crops | 95 | 89 | 748 | 176 | 317 | 45 | 1470.02 | 65 | 182 | 0.3 | 16.3 | 54.7 | 4.79 | 12.5 |
| Cotton | 8152 | 7561 | 16,314 | 13,078 | 6169 | 4160 | 55,431.8 | 221 | 72 | 0.8 | 55.3 | 21.5 | 341.3 | 103.0 |
| Sub-total (summer crops), ton/year | | | | | | | | | | | | | 5222.5 | 1742.0 |
| Winter crop | Terah | Drain 7 | Drain 8 | Drain 9 | Drain 11 | Gharbia | Total area, feddan | Nf, kg/ha | Pf, kg/ha | L, m | Nu | Pu | TN total, ton | TP total, ton |
| Long clover | 9597 | 12,209 | 43,982 | 13,262 | 19,266 | 11843.0 | 110,160 | 144 | 91 | 0.2 | 35.9 | 27.4 | 1068 | 633 |
| Wheat | 19,307 | 32,321 | 82,705 | 44,059 | 38,729 | 22427.3 | 239,548 | 234 | 52 | 0.4 | 58.6 | 15.6 | 2517 | 522 |
| Barley | 33 | 307 | 224 | 19 | 855 | 34.9 | 1474 | 59 | 20 | 0.4 | 14.7 | 5.9 | 3.6 | 1.1 |
| Broad bean | 6975 | 3421 | 7895 | 3538 | 4594 | 4217.6 | 30,641 | 98 | 250 | 0.5 | 24.5 | 75.1 | 108 | 256 |
| Lentil | 0 | 0.3 | 0 | 0 | 0 | 1.35 | 2 | 234 | 52 | 0.6 | 58.6 | 15.6 | 0.012 | 0.003 |
| Flax | 202 | 318 | 327 | 130 | 47 | 407.2 | 1431 | 80 | 45 | 1.2 | 20.1 | 13.6 | 2.7 | 1.4 |

**Table 4.** *Cont.*

| Summer Crop | Terah | Drain 7 | Drain 8 | Drain 9 | Drain 11 | Gharbia | Total Area, Feddan | Nf, kg/ha (Fertilizers + FYM) | Pf, kg/ha (Fertilizers + FYM) | L, m | Nu | Pu | TN Total, ton | TP Total, ton |
|---|---|---|---|---|---|---|---|---|---|---|---|---|---|---|
| Onion—winter | 3 | 1 | 14,360 | 54 | 3 | 1347.1 | 15,768 | 189 | 52 | 0.3 | 47.2 | 15.6 | 148.6 | 38.3 |
| Garlic | 0 | 0 | 70 | 0 | 0 | 2.24 | 72 | 143 | 52 | 0.1 | 35.8 | 15.6 | 1.1 | 0.4 |
| Sugar beet | 16,121 | 15,189 | 13,550 | 8445 | 5319 | 4982.1 | 63,607 | 78 | 104 | 0.9 | 19.6 | 31.3 | 131.5 | 162.9 |
| Potato | 32 | 56 | 2928 | 120 | 106 | 886.4 | 4129 | 390 | 189 | 0.5 | 97.6 | 56.7 | 57.7 | 26.0 |
| Tomato | 155 | 39 | 31 | 20 | 427 | 35.2 | 706 | 390 | 143 | 0.5 | 97.6 | 43.0 | 9.9 | 3.4 |
| Other vegetables | 412 | 320 | 3719 | 500 | 627 | 660.6 | 6239 | 325 | 280 | 0.3 | 81.4 | 84.0 | 101.3 | 81.2 |
| Tahrish | 6430 | 4918 | 8852 | 4503 | 2207 | 2266.6 | 29,177 | 390 | 189 | 0.5 | 97.6 | 56.7 | 407.5 | 183.7 |
| Other crops | 4 | 1 | 861 | 2 | 0 | 105.1 | 973 | 65 | 182 | 0.3 | 16.3 | 54.7 | 3.2 | 8.3 |
| Sub-total (winter crops), ton/year | | | | | | | | | | | | | 4559.6 | 1917.5 |
| Total | | | | | | | | | | | | | 9782.0 | 3659.5 |

**Table 5.** WWTPs discharging their effluents to agricultural drains leading to Lake Burullus (data obtained from Holding Company for Water and Wastewater (HCWW)).

| No. | Final Disposal Ooutlet | Governorate | WWTP | Q * mm$^3$/Year |
|---|---|---|---|---|
| 1 | Tira drain | Kafr El Sheikh | Baltim area | 0.73 |
| 2 | Burullus Lake | | Burullus | 1.752 |
| 3 | Drain no. 11 | | Fowa | 1.46 |
| 4 | | | Motobas | 0.73 |
| 5 | Drain no. 7 | | El Ryad | 0.73 |
| 6 | | | Sakha | 14.6 |
| 7 | | Gharbia | Basyoun | 2.19 |
| 8 | | | Kafer El Zayat | 14.6 |
| 9 | | Kafr El Sheikh | Qaleen | 1.825 |
| 10 | | | Desouk | 1.825 |
| 11 | | | Al Agouzen | 0 ** |
| 12 | | | Sedi Salem | 0.73 |
| 13 | Gharbia drain | Gharbia | Tanta (first stage) | 21.9 |
| 14 | | | Tanta (second stage) | 10.95 |
| 15 | | | Nawag | 0.5475 |
| 16 | | | Zeyad | 2.555 |
| 17 | | | Mahala Kobra | 16.425 |
| 18 | | | El Matemadia | 0.5475 |
| 19 | | | Beshbesh | 0.511 |
| 20 | | | Saft Trab | 1.46 |
| 21 | | | Kotor | 0.365 |
| 22 | Burullus Lake | | Mahalet Marhoum | 1.6425 |
| 23 | Gharbia drain | Kafr El Sheikh | Biala | 0.73 |
| 24 | | | Hamool | 1.46 |
| | Total | | | 59.09 |

\* Q = Actual measured outflow of effluents. \*\* No available measured outflow of effluents.

**Table 6.** Flow and pollution loads into Lake Burullus due to domestic wastewater pollution component.

| Item | | | Drains | | | | | | | Total to LB |
|---|---|---|---|---|---|---|---|---|---|---|
| | | | Terah | Burullus 15% Gharbia | Drain 7 | Drain 8 | Drain 9 | Drain 11 | Lake Burullus | |
| Flow, million m$^3$/year | Total flow | | 161.9 | 137.7 | 206.7 | 408.3 | 225.7 | 185.3 | 1.8 | 1327.5 |
| | Domestic flow | Direct | 0.7 | 8.9 | 15.3 | | 21.2 | 2.2 | 1.8 | 50.0 |
| Population within catchment area | Served with sanitation | | 948,381 | 113,015 | 93,663 | 241,791 | 366,343 | 121,135 | | 1,884,327 |
| | Unserved with sanitation | | 118,536 | 234,208 | 238,994 | 768,140 | 392,621 | 332,798 | | 2,085,296 |
| | Total | | 1,066,917 | 1,066,917 | 332,656 | 1,009,930 | 758,965 | 453,933 | | 3,969,624 |
| COD pollution load, ton/year | Direct | Worst | 219 | 2659 | 4599 | | 6351 | 657 | 526 | 15,011 |
| | | Best | 58 | 709 | 1226 | | 1694 | 175 | 140 | 4003 |
| | Indirect | | 2667 | 5270 | 5377 | 17,283 | 8834 | 7488 | | 46,919 |
| | Total with best scenario | | 2725 | 5979 | 6604 | 17,283 | 10,528 | 7663 | 140 | 50,922 |
| | Total with worst scenario | | 2886 | 7929 | 9976 | 17,283 | 15,185 | 8145 | 526 | 61,930 |
| TN pollution load, ton/year | | | 266 | 823 | 1091 | 1536 | 1632 | 753 | 70 | 6172 |
| TP pollution load, ton/year | | | 48 | 132 | 166 | 288 | 253 | 136 | 9 | 1032 |

Data reported from different sources on the actual acreage and fish production from fish farms around Lake Burullus deviate. Three main sources were considered to obtain such data, namely, Ministerial Decree no. 329/1985, the EEAA, and the General Authority for Fish Resources Development (GAFRD). Ministerial Decree no. 329/1985 allocated an area of 27,512 acres (111.34 km$^2$) around Lake Burullus to fish aquaculture. In its 2008 report on the environmental condition of the Kafr El Sheikh governorate, the EEAA noted that the acreage of fish farms around Lake Burullus amounted to 35,254 acres (142.67 km$^2$) with a production capacity of 67,170 tons/year. In a more recent report on the current situation and proposals for the development of the lake (July 2011), the EEAA indicated that fish farms at the southern parts of Lake Burullus extended over an area of 29,636 acres (119.93 km$^2$) with a production capacity of 181,000 tons/year and an annual revenue of 1,629,000,000 LE, using an average price of 9 LE per kg. Of these 29,636 acres (119.93 km$^2$), 321 fish farms leased on the lake by the Fisheries Authority covered an area of 4636 acres (18.76 km$^2$), while the area of farms leased as property of the state around the lake was 25,000 acres (101.17 km$^2$). Finally, according to GAFRD data [69], the estimated area of fish farms around Lake Burullus was 99,280 acres (401.76 km$^2$) with a production capacity of 297,557 tons/year.

The cultivation system for fish aquaculture in fish farms (semi-intensive aquaculture) around Lake Burullus mainly depends on building an earthen dike of very low permeability along the perimeter of the fish farm that allows the creation of a water basin with a water depth ranging from 1.2–1.5 m. The basin is then filled with water from the nearest agriculture drain via a diesel water pump. The fry is the seed to the system, and feeder is added gradually to the basin according to the growth rate of the fish (10–35,000 fish/acre). The types of cultivated fish are tilapia Nile and mullet Nile. Natural aeration of the basin is commonly applied; water is replaced via two diesel water pumps every two weeks in order to avoid accumulation of fish feces (COD, TN, and TP). Availability of water in terms of quantity and quality during the cultivation cycle (8–10 months) is the single most important factor affecting fish production and related waste generation of these farms. Yet, the only water source for feeding the fishing system is the agriculture drains near the fish farms. Thus, most of the fish farms suffer from their ultimate need to change the water in their basins during periods of fish growth, which may coincide with periods of shortage and poor water quality in the agriculture drains, especially during summer periods. The situation worsens in the case of fish farms located at the tail-ends of the agriculture drains close to Lake Burullus, which may receive the wastewater output of more upstream farms (upstream-downstream drainage recirculation). Such a situation might lead to an increase in the concentration of fish feces and a reduction in oxygen concentration that negatively affects cultivation efficiency.

The amount of pollutants produced by fish farm waste was estimated on the basis of the average quantity of feed used (6 tons/acre during the production cycle), the feed conversion ratio of 0.5 fish weight/feed weight, and the reported composition of fish feces under similar conditions. Table 7 shows that the maximum COD, TN, and TP loads were about 50,700, 11,250, and 2800 tons/year, respectively.

**Table 7.** Pollution load into Lake Burullus due to fish aquacultural component.

| Source | Fish Farm Area (acres) | Fish Production (ton/year) | Pollutants (ton/year) | | |
|---|---|---|---|---|---|
| | | | COD | TN | TP |
| [70] | 35,254 | 67,170 | 28,500 | 6350 | 1590 |
| [69] | 99,280 | 297,557 | 50,700 | 11,250 | 2800 |

### 3.2.5. Solid Waste Pollution Component

Due to the limited number of sanitary landfill sites around the country, solid waste is most frequently disposed of in uncontrolled public landfills or dump sites. Thus, leachate from its degradation typically dissipates into the ground or drains to the nearest natural

receptor. This situation is more severe in rural Egypt. Due to the inefficiency of the solid waste collection/transfer system in the catchment basin of the agricultural drains leading to Lake Burullus, solid waste generated in the villages along these drains is dumped close to the drains, on the drain banks, or within the drains themselves, inducing serious environmental impacts. Although the preceding remarks characterize a recurring field situation, quantitative data have not been systematically collected to allow for a reliable estimate of the extent and intensity of the issue. As a result, it has not been feasible to assess the pollution load of COD, TN, and TP resulting from solid waste entering the drainage network, and consequently, the lake, due to the difficulty of estimating their values from the available data.

### 3.2.6. Summary of Pollution Loads Discharged to Lake Burullus

Table 8 summarizes the pollution loads to Lake Burullus emanating from its main pollution components, i.e., (a) the agriculture pollution component, (b) the domestic wastewater (sanitary) component, (c) the industrial wastewater pollution component, and (d) the fish aquaculture (fish farm) pollution component. The highest organic load is due to the domestic wastewater pollution component, which represents 63.2% of the total COD pollutant load to Lake Burullus. The results also show that the agricultural pollution component is the major contributor to the TP (58.3%) and TN (43.9%) pollutant loads to Lake Burullus. The industrial wastewater pollution component has a very small effect on the pollution of Lake Burullus.

**Table 8.** Summary of pollution loads into Lake Burullus due to main pollution components.

| Pollutant | Pollution Load to Lake Burullus | | | | | | | | Total Load (Calculated), ton/year | Total Load (Measured *), ton/year | Measuring Year |
|---|---|---|---|---|---|---|---|---|---|---|---|
| | Agriculture | | Domestic | | Industrial | | Fishery | | | | |
| | Load, ton/year | % | Load, ton/year | % | Load, ton/year | % | Load, ton/year | % | | | |
| COD load (ton/year) | | | 50,922 | 63.2 | 1163 | 1.4 | 28,500 | 35.4 | 80,585 | 82,400 | 2014–2015 |
| TN load (ton/year) | 9782 | 43.9 | 6172 | 27.7 | | | 6350 | 28.5 | 22,304 | 24,000 | 2014–2015 |
| TP load (ton/year) | 3660 | 58.3 | 1032 | 16.4 | | | 1590 | 25.3 | 6281 | 4600 | 2014–2015 |

* Measured by Ministry of Water Resources and Irrigation [65].

Table 8 further presents a comparison between (a) the total COD, TN, and TP pollutant loads estimated to discharge to Lake Burullus, and (b) the respective COD, TN, and TP pollutant loads measured at the pumping stations located at the tail-ends of the agricultural drains discharging to Lake Burullus. The variation between the calculated and measured values for COD and TN is not significant, but the variation for TP is significant. This may be attributed to the characteristics of the soil in the study area, where a high soil pH (ranges from 8.1 to 8.5) encourages the precipitation of phosphate with calcium carbonate [71–74]. Further research is needed to include the characteristics of the soil as a main variable in estimating the TP load to the lake. It is also worth mentioning that the main polluters to the lake are COD and TN, where high COD in the lake causes dissolved oxygen depletion, while high TN causes massive algae growth [75–78]. The variance noted in these results may also be attributed to the following subsequent reasons:

- The approximate approach taken to the estimation of COD, TN, and TP pollutant loads. A more refined study may identify how the COD, TN, and TP concentrations change from the upstream inlets to the downstream outlets of the agricultural drains.
- The reuse of drainage water in agriculture (indirect use) to utilize the available TN and TP nutrients in it, and thus, compensate for the additional quantities of fertilizers needed (the quantities of fertilizers sold in the Kafr El Sheikh governorate are much lower than those actually necessary due to the dependency of farmers on the use of sludge from WWTPs as compost and the reuse of drainage water to take benefit of the

TN and TP elements in it; these elements are in a high concentrations due to domestic wastewater arriving partially treated or untreated into agricultural drains).

- The growth of water plants (e.g., reeds) and algae in the agricultural drainage network that feed on the TN and TP nutrients available in it. These plants grow throughout the year but more intensively during summer with the rise in temperature.
- Sedimentation along the agricultural drainage network due to low water velocity in the drainage section, as agricultural drains often represent a type of primary settlement for untreated domestic wastewater (self-purification).
- The lack of flow and COD, TN, and TP concentration measurements from branch drains to main drains and their outlets to Lake Burullus in order to track degeneration of these pollutants through self-purification or reuse (resource recovery).
- The lack of continuous measurements throughout the year over consecutive years to track the seasonal and temporal variations of the COD, TN, and TP pollutants.

## 4. Conclusions and Further Research

For the case of Lake Burullus, we conclude that the domestic wastewater pollution component generates the highest organic load (63.2% of the total COD pollutant load to Lake Burullus). Agriculture is the main contributor of nutrient loading to Lake Burullus, where it represents 43.9% of the TN pollutant load and 58.3% of the TP pollutant load to the lake. Fish aquaculture is the second main contributor after agriculture, resulting in about 28.5% of the TN load and 25.3% of the TP load. The industrial wastewater pollution component has a very small effect on the pollution of Lake Burullus. The proposed methodology has succeeded in assessing the pollution loads in terms of COD, TN, and TP for the main human-based activities (agriculture, industrial, domestic and urban development business, fish farming, etc.). The findings allow decision-makers to prioritize the interventions and investments that will achieve their target and meet the countrywide vision. Additionally, the proposed pragmatic approach can be used for continuous monitoring and evaluation of human-based activities, and at the same time, to follow-up the development of projects' impact after their completion. This approach needs further development by continuous monitoring of pollution loads at the hotspots that have been identified, to continuously evaluate the impact of human-based activities, as well as to evaluate the impact of implemented/proposed interventions for pollution reduction for the main sectors involved. Consequently, the proposed approach can be employed as a knowledge-based decision system for decision-makers.

**Supplementary Materials:** The following are available online at https://www.mdpi.com/article/10.3390/w13233337/s1, Table S1: Average root depth for various summer and winter crops (in m), Table S2: Annual quantities of fertilizers for various crops (Kg/hectare/year), Table S3: Annual quantities of FYM for various crops (Kg/hectare/year).

**Author Contributions:** Data curation, M.K.M.; conceptualization, formal analysis, investigation, methodology, project administration, resources, supervision, validation, visualization, writing—original draft, review, and editing, M.S.M. and M.K.M. All authors have read and agreed to the published version of the manuscript.

**Funding:** The authors received no financial support for the research, authorship, and/or publication of this article.

**Institutional Review Board Statement:** Not applicable.

**Informed Consent Statement:** Not applicable.

**Data Availability Statement:** Not applicable.

**Acknowledgments:** The authors would like to express their deepest gratitude to the Ministry of Irrigation and Water Resources (MIWR), Egyptian Environmental Affairs Agency (EEAA), Ministry of Agriculture and Land Reclamation (MALR), and the Holding Company for Water and Wastewater (HCWW) for their technical support with this study.

**Conflicts of Interest:** The authors declare no conflict of interest.

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
