# Peer review of "Rapid Assessment Method for Evaluation of the Weighted Contribution of Anthropogenic Pollution: A Case Study of Lake Burullus, Egypt"

_water, doi:10.3390/w13233337_

Round 1
Reviewer 1 Report
In the manuscript, “A methodology for rapid assessment of the weighted contribution of human based activities to lake pollution: A case study of Lake Burullus, Egypt”, the impact of the anthropogenic pollutant to the water quality in the lake was assessed.
I could not find the novelty in this study at all. The results are not well discussed and conclusions are not meaningful in manuscript. I think this study is just the local report or the introduction of the Lake Burullus. Even if this study could be categorized as “the case study”, only showing the results was not satisfied. At least, comparing to other lakes in Egypt or in the world were required. Furthermore, I find the overall writing, both grammar and content, to be poor. I would suggest a significant edit be made to correct these errors.
Author Response
We appreciate the time and effort that you dedicated to providing feedback on our manuscript and are grateful for the insightful comments on and valuable improvements to our paper. We have incorporated most of the suggestions made by you. Those changes are highlighted within the manuscript. Please see below, in red, for a point-by-point response to your’ comments and concerns.

Reviewer 2 Report
I thoroughly reviewed this paper and found it to be suitable for the journal. However, the authors need to overcome some of the major deficiencies
indicated below;
• The abstract is a bit complicated and wage for my likings. I recommend authors to make it easy to read, concise, and comprehensive.
• Although I’m not a native English speaker and the language used in manuscript is mostly of high quality, but I noticed a lot of punctuations and structural errors. I would recommend authors to get their manuscript reviewed by a native English reviewer.
• The hypothesis of the manuscript is unclear. The authors are advised to clearly write the hypothesis of their work in the context of shortfalls of the previous research works.
• While the authors rewrite the hypothesis of their work, they must explicitly include the novelty of their work/ methodology.
• In Result and Discussion Section, the authors must statistically indicate that the hypothesis of their work is proved.
Author Response

(The authors gave the same response as above.)

Reviewer 3 Report
Dear Authors,
Please see the section-wise comments below on your research article with the following details.
Manuscript title: A methodology for rapid assessment of the weighted contribution of human based activities to lake pollution: A case study of Lake Burullus, Egypt
Manuscript Number: water-1420562
Journal Submitted: Water
Specific Comments:
This manuscript has a severe problem with the English language and description. Please revise it with the help of an expert in this field.
Another general suggestion is that this study has been conducted as a local level study that seems very hard to relate to the different parts of the world. If you continue with the same consistency, I am afraid this paper may not be suitable for an international journal like Water.
Title:
The title can be modified as “Rapid assessment method for evaluation of the weighted contribution of anthropogenic pollution: A case study of Lake Burullus, Egypt”
Abstract:
L 13: Practical and pragmatic are the same thing.
L 27: COD pollutant?
This is not abstract. This is a summary of your methods.
Please revise it whole and add results rather than methods.
There are conclusions.
L 14-16: Please move this at the end and modify it according to your conclusions.
Keywords: Please revise the keywords and especially “quick pollution assessment”.
Introduction:
This introduction is in fact a summary of the study area which is not a suitable idea. You may dedicate a paragraph about the study area in the methods. However, there is a severe lack of background literature on the weight contribution of pollutants. Therefore, the whole introduction should be revised.
Please make sure that you clearly define your objectives.
Materials and Methods:
L 112-131: There is nothing significant mentioned here. Please delete and concentrate upon what has actually been done during this study.
This section must be revised.
Please explain your study design, area first and then provide the other details.
I could not find the statistical details.
Please cite the relevant references as well because I could not find many background citations.
Results and Discussion:
L 303-312: Are these not your own findings?
I guess you have missed the critical links here.
I can see that the results are described very well. However, the discussions are too poor.
It would help if you made the results more precise and understandable for the other researchers.
I suggest adding some figures in the results section as well.
While writing the discussions again, it would be best to discuss the results from multiple angles and placed them into context without being overinterpreted.
Figures and Tables:
The tables are complex and too long. Please find common points and try to decrease their content.
Conclusions
The conclusions are okay. However, they will be revised upon revisions in the introduction, methods, and discussions parts.
References:
More than 80% of the references are local. Please add other citations as well.
Author Response

(The authors gave the same response as above.)

Round 2
Reviewer 3 Report
Dear Authors,
Thanks for providing the revised version. I can see that you have done a great job while revising this manuscript. I have two concerns now.
Please delete the word "Pollutant" from the x-axis of figure 2.
Please merge the "future perspective" section into the conclusions and change it to "Conclusions and further research".
The rest appears to be fine to me.
Author Response
Dear Reviewer,
We appreciate the time and effort that you dedicated to providing feedback on our manuscript and are grateful for the insightful comments on and valuable improvements to our paper. We have addressed the two comments you sent to us. Thanks again for your support.
Regards,
Authors of this manuscript
